# HemaRAG: A Retrieval-Augmented Generation System for Medical Question Answering in Hematologic Malignancies*

Maria Evangelia Chatzimina
Dept. of Electrical and
Computer Engineering,
Hellenic Mediterranean
University & Institute of
Computer Science, Foundation
for Research and Technology–
Hellas (FORTH) Heraklion,
Crete, Greece
hatzimin@ics.forth.gr

Nikolaos S. Tachos
Unit of Medical Technology
and Intelligent Information
Systems Department of
Materials Science and
Engineering, University of
Ioannina Ioannina, Greece
ntachos@gmail.com

Dimitrios I. Fotiadis
Unit of Medical Technology
and Intelligent Information
Systems, Department of
Materials Science and
Engineering, University of
Ioannina, Ioannina, GR 45110
Greece, and the Department of
Biomedical Research Institute,
FORTH, Ioannina, Greece
fotiadis@uoi.gr

Manolis Tsiknakis
Dept. of Electrical and
Computer Engineering,
Hellenic Mediterranean
University & Institute of
Computer Science, Foundation
for Research and Technology–
Hellas (FORTH)
Heraklion, Crete, Greece
tsiknaki@ics.forth.gr

*Abstract*— **Answering complex medical questions requires both reliable information retrieval and the ability to generate responses that are medically accurate and contextually appropriate. In this paper, we present HemaRAG, a Retrieval-Augmented Generation (RAG) system designed specifically for hematologic malignancies. Our system combines a dense retriever enhanced with biomedical ontologies and a fine-tuned large language model (Gemma 3), trained locally on domain-specific literature and question–answer pairs. To build a robust retrieval base, we enriched PubMed abstracts and curated datasets such as BioASQ and PubMedQA using synonym mappings from MeSH, NCIT, DOID, and UMLS. We used a local vector database to support high-speed semantic search without sharing data externally. Evaluation across both BioASQ and long-form PubMedQA benchmarks showed high semantic accuracy (BERTScore: 87–89%), strong lexical overlap (F1: 49–52%), and high retrieval performance (Recall@10: 94–96%), despite the challenges posed by free-form medical questions. The system was developed and deployed entirely locally making it suitable for clinical contexts where patient data privacy is essential. In future work, we plan to integrate HemaRAG into an empathetic conversational agent designed to support patients and clinicians in the field of hematologic oncology.**

*Keywords*— *Retrieval-Augmented Generation, RAG, LLMs, Question Answering, PubMed, BioASQ, Clinical NLP, Hematology, Hematologic Malignancies*

## I. INTRODUCTION

Large language models (LLMs) have made major progress in understanding and generating natural language. LLMs show promising results on general tasks like summarization, dialogue, and open-domain question answering (QA) in general and medical domain [1][2][3][4][5]. Although recent studies highlighted the performance of LLMs, they often generate factually incorrect responses which is called hallucinations [6]. Moreover, in specialized areas like medicine and especially in hematologic malignancies LLMs often struggle. Important factor for poor results in specialized domains is the lack of focused and high quality data. Most LLMs are trained on general and public text sources. These lack the detailed, technical, and often subtle language used in medical research and clinical care. As a result, even advanced LLMs can struggle to understand medical terminology or make accurate inferences when answering questions about blood cancers like leukemia or lymphoma. To address these problems, Retrieval-Augmented Generation (RAG) has emerged as a promising solution. By retrieving relevant documents from reliable and up-to-date sources and presenting them to the LLM during generation, RAG can help mitigate hallucinations and avoid outdated information [7][8].

Hematologic malignancies such as leukemia, lymphoma, and multiple myeloma affect hundreds of thousands of patients globally and often require ongoing care, particularly in advanced stages [9]. Since hematologic malignancies are chronic and often life-limiting with psychological and physical burden, palliative care is an important factor in improving the patients' quality of life. In palliative settings, patients and caregivers seek reliable information in order to manage symptoms, understand treatment options and make informed decisions. In these sensitive contexts, even minor misunderstandings can lead to distress or poor outcomes. This makes the need for accurate, domain-specific language understanding and documents retrieval crucial.

While RAG systems offer a promising way to address this gap by retrieving relevant texts before generating an answer, they also face challenges. Many RAG pipelines rely on simple keyword matching or generic embeddings, which show low performance with the complex, synonym-rich language of medicine. For example, a question mentioning "AML" might not match an article that only uses "acute myeloid leukemia," leading to poor retrieval and wrong answers.

Our work builds on prior efforts in medical QA [10], domain-specific LLMs [11], and RAG in clinical settings [12][13], while focusing on a specific cancer domain and including biomedical enrichment to improve retrieval quality. The goal of our research is to address this gap by implementing HemaRAG, a domain-specific RAG system designed to support medical QA in hematologic malignancies. Our system combines a PubMed collection of articles focused on blood cancers, enrichment using biomedical ontologies like MeSH, NCIT, and DOID to add synonyms and concept definitions and evaluation using the BioASQ and PubMedQA datasets.

This enriched retrieval process allows the system to bridge lexical gaps and improve semantic coverage by helping the model retrieve and reason over the most relevant information. Our system uses a finetuned Gemma 3 27 billion parameters

model [14] with data focused on hematologic malignancies and the retrieved passages are given directly into its input, improving its performance in a transparent and efficient way.

Our results show that this method improves answer quality and supports more accurate medical reasoning. By combining enriched biomedical retrieval with a domain finetuned generative model, HemaRAG demonstrates how domain-specific RAG systems can be adapted to support real-world medical tasks

## II. RELATED WORK

RAG is a promising method for grounding LLMs in external domain-specific knowledge, particularly in the healthcare domain. As LLMs like GPT-3.5 and GPT-4 gain popularity, researchers have explored how RAG can be used to improve factual accuracy, mitigate hallucinations, and support clinical reasoning. Despite encouraging results, many existing systems rely on closed, API-based models [15], raising concerns about transparency, reproducibility and data privacy. Moreover, most health-related RAG systems have been developed for general clinical contexts, with few focusing on narrow medical specialties.

Recent work has introduced increasingly sophisticated retrieval mechanisms to address the limitations of naive RAG pipelines. Iterative RAG systems such as iMedRAG [16] refine their retrieval using follow-up queries and demonstrate the potential of iterative querying in healthcare applications. Other researches have expanded RAG into multimodal and structured data domains. MMed-RAG [17] incorporates both medical images and text which enhance the system's ability to answer visual medical questions with grounded evidence. Similarly, RGAR [18] retrieves a combination of factual and conceptual knowledge from clinical text and electronic health records, enabling models to outperform even larger LLMs like GPT3.5 in fact-sensitive medical QA. Dialogue-based systems such as MRD-RAG [19] have also been proposed, using multi-turn conversational setups to better simulate diagnostic reasoning through follow-up exchanges. Another research focuses on fact injection by retrieving medical data from a disease database and incorporates them into the model's prompt [20]. To our knowledge, the only work that applies a RAG-style chatbot specifically to hematologic malignancies is focused on multiple myeloma and even though it shows the feasibility of RAG in this area, its scope is limited and cannot be generalized across other hematologic cancers [21].

Most systems described above are using external models and services which can transit sensitive clinical information and queries to external databases and servers. Patient privacy and data protection of sensitive clinical data is an important part of medical tools. A recent review noted that while retrieval-based methods are increasingly used in healthcare, few implementations explicitly address ethical concerns or regulatory constraints, despite using data sources such as PubMed, clinical guidelines, or even EHRs [22].

Our system, HemaRAG, is designed to be deployed entirely on local infrastructure. All model inference and document retrieval are performed without any external API calls, making the system suitable for privacy-sensitive environments such as hospital research networks or palliative care settings. Moreover, HemaRAG focuses on hematologic malignancies. These diseases often involve terminology that varies significantly across articles, clinicians, and countries making semantic retrieval especially challenging. To address this, our pipeline includes enrichment with biomedical ontologies such as MeSH, NCIT, and DOID and UMLS. This enrichment is applied both to the retrieved corpus and to user questions, improving semantic overlap and retrieval performance. To our knowledge, HemaRAG is the first retrieval-augmented system built specifically for hematologic cancer QA. Its combination of local execution, ontology-driven enrichment, and domain-specific focus addresses a real gap in the medical RAG research domain.

## III. METHODS

This section presents the architecture and implementation of HemaRAG, a RAG system designed for medical QA in the domain of hematologic malignancies. The pipeline operates entirely on local infrastructure, and consists of five key components: (1) domain-specific document collection, (2) biomedical enrichment, (3) embedding and indexing with Chroma DB [23], (4) LLM-based generation, and (5) evaluation using external benchmarks. In the architecture of the system is shown Fig. 1.

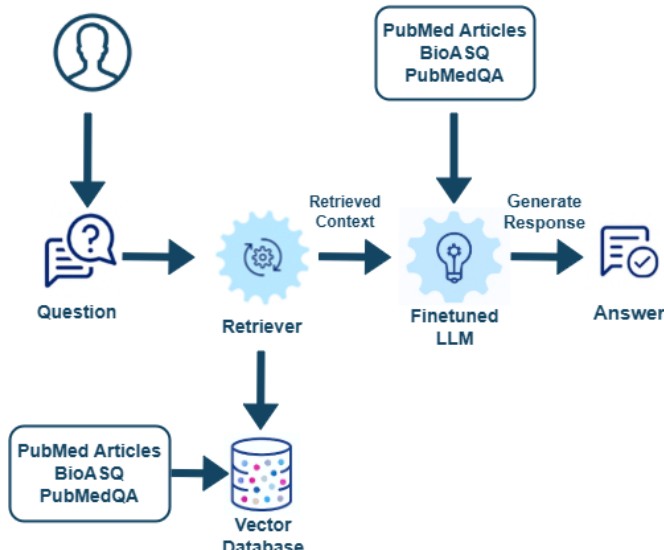

Fig. 1: RAG Architecture

### A. Corpus Collection from Biomedical Databases

A domain-specific corpus was constructed by querying both PubMed and Europe PMC using their respective public APIs. In order to collect wide range of relevant literature of hematologic malignancies, a comprehensive query was created that included disease names (e.g., "acute lymphoblastic leukemia," "Waldenström macroglobulinemia"), treatments (e.g., "CAR-T," "venetoclax," "daratumumab"), and clinical concepts (e.g., "minimal residual disease," "bone marrow transplant"). Filtering based on relevance (based on keyword matches in titles and abstracts) was applied and the results from both sources were merged. The pipeline also handled full-text extraction where available via the Europe PMC XML interface. Articles without sufficient domain relevance were excluded.

Finally, two additional biomedical QA datasets were included in our corpus: BioASQ [24] and the long-form version of PubMedQA [25]. BioASQ is a benchmark dataset designed to support open-domain biomedical QA, providing manually curated question–answer pairs along with

supporting documents. It includes factoid, list, and summary-style answers written by biomedical experts, making it a strong foundation for fine-tuning generative models in the medical domain. The long-form version of PubMedQA was also included, which contains clinical research questions paired with full-sentence answers. Unlike the standard version that provides yes, no and maybe labels, the long-form variant provides full-sentence answers that are based on actual PubMed abstracts. This format aligned with our system, which aims to generate semantically faithful, evidence-grounded responses rather than binary labels.

*B. Ontology-Based Biomedical Enrichment*

Biomedical terminology is highly variable, with diseases and treatments often referenced by multiple names or abbreviations. To bridge this lexical gap during both indexing and retrieval, we enriched all documents using biomedical ontologies. Specifically, we queried BioPortal [26] to extract synonyms and preferred terms from MeSH, NCIT, and DOID [27][28] ontologies. Terms like "CLL," "chronic lymphocytic leukemia," and "B-cell malignancy" were normalized and mapped to shared concept labels.

To enhance semantic grounding, we also parsed the UMLS Metathesaurus [29] locally, allowing term-to-Concept-level linking (CUIs) mapping and synonym expansion beyond BioPortal coverage. This expansion was used during preprocessing and retrieval to improve matching between question phrasing and literature content. We applied synonym enrichment conservatively to avoid introducing errors. Only exact or high-confidence matches (e.g., acronyms, preferred terms) were used, and we manually reviewed samples of these to ensure correctness. This review was performed iteratively in order to ensure alignment between enriched queries and retrieved content. This enrichment strategy was applied to both the question inputs and the documents, helping to improve the alignment between the two during retrieval. In fig. 2 the ontology enrichment pipeline is shown.

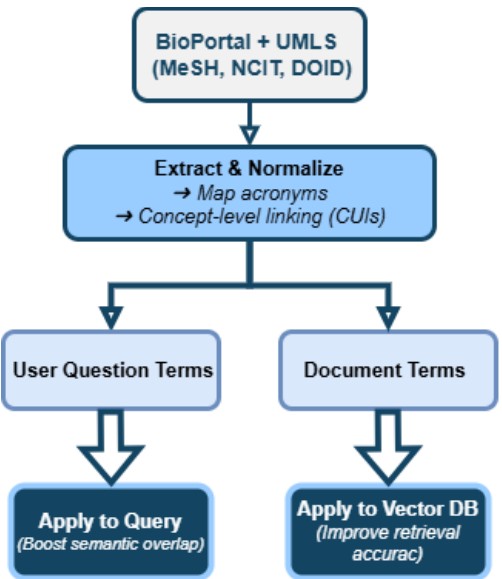

Fig. 2: Biomedical Enrichment Pipeline

*C. Embedding and Indexing with ChromaDB*

To manage our data storage and retrieval Chroma DB was used which is a lightweight and local vector database. Each document in was split into smaller text chunks of 1,000 characters and 20% overlap, in order to make indexing and retrieval more efficient. Metadata was added to each chunk, including the original question (when available), PubMed ID, and, for BioASQ and PubMedQA the correct answer. This extra layer helped us later, especially when debugging retrieval results or doing error analysis.

Based on the research of Xiong et al. benchmark which showed that dense retrievers outperform traditional keyword-based retrieval methods in medical QA tasks, we used a dense embedding model for all retrieval operations [13] [30]. Embeddings were stored and queried using Chroma DB, enabling semantic retrieval from PubMed, BioASQ, and PubMedQA documents. Embeddings were generated using GPU acceleration and normalized for cosine similarity. The test sets of BioASQ and PubMedQA were excluded from storing in the vector database.

*D. Large Language Model Fine-Tuning*

For response generation, we used the Gemma 3 27B model, hosted and fine-tuned entirely on local infrastructure. The model was trained on domain-specific abstracts from PubMed, PubMedQA long form and question–answer pairs from the BioASQ dataset. To preserve evaluation fairness, all BioASQ and PubMedQA test items were excluded from training and retrieval.

For the evaluation, we filtered 500 hematologic cancer-related questions from PubMedQA and randomly selected 30% (150 questions) for testing. Similarly, from the 378 questions collected from BioASQ after disease-specific filtering, we used 113 (30%) for evaluation. The remaining samples were used for the finetuning.

Fine-tuning followed a two-stage strategy. First, the model was trained to the domain-specific biomedical corpus (comprising 12573 retrieved abstracts and full texts) using a masked token prediction objective, allowing it to adapt to medical terminology and phrasing. Moreover, 11573 samples were used for the training and 1000 for validation. In the second stage, the model was fine-tuned on 90% of the question–answer pairs from the filtered BioASQ and PubMedQA sets, using instruction-style formatting. The remaining 10% was reserved for validation. This stage aimed to improve factual accuracy, answer structure, and domain-specific reasoning.

We selected Gemma 3 27 billion parameters because it combines strong instruction-following ability with the flexibility needed for local deployment. The model is open-source, which made it easier to integrate into our pipeline and fine-tune on domain-specific data without relying on external services. It also supports efficient training through 4-bit quantization and gradient checkpointing, allowing us to work with large models on limited hardware. Moreover, its performance in LLMs benchmarks suggests that it could be a valuable tool in health and medical tasks.

Training was performed using 4-bit quantization with gradient checkpointing and mixed precision to minimize memory footprint. We used a batch size of 2, gradient accumulation steps of 4, and the AdamW optimizer with a learning rate of 1e-4. Training on PubMed was conducted for 200 steps, while BioASQ fine-tuning lasted 300 steps. These values were selected based on dataset size and the stabilization of loss and perplexity curves during training. Although BioASQ is smaller, it required more steps due to its

complexity and structure. Finetuning was conducted with an NVIDIA A40 GPU (48 GB) and completed in around 9 hours. To monitor training quality, we computed perplexity on held-out validation splits from both BioASQ and PubMed. The model achieved a final perplexity of 19.28 on BioASQ and 14.42 on PubMed, indicating improved domain adaptation.

*E. Retrieval-Augmented Generation*

User questions are enriched using ontology mappings, embedded using the same embeddings model and passed to Chroma DB for top-k retrieval. Retrieved passages are then formatted alongside the query in a structured prompt and fed to the fine-tuned Gemma model. We evaluated HemaRAG using two gold-standard benchmarks: BioASQ and PubMedQA. For both benchmarks, we used a held-out test set extracted prior to fine-tuning. The system-generated answers were evaluated against reference answers using:

For both BioASQ and PubMedQA evaluations, we used a two-stage pipeline. In the retrieval phase, we computed Recall@K metrics (K=5, 10) to assess whether the top-k documents included relevant evidence. Our retriever combined dense embeddings and cross-encoder reranking, followed by a diversity-aware final ranking.

In the generation phase, we evaluated system answers against gold references using standard metrics including Exact Match, F1-score, ROUGE-L[31], BERTScore [32], and Named Entity Recognition (NER) overlap. ROUGE-L measures the longest common subsequence between generated and reference texts, capturing fluency and overlap. BERTScore computes similarity based on contextual embeddings from a pretrained biomedical language model, allowing for a more accurate reflection of semantic alignment between model outputs and reference answers.

## IV. RESULTS

We evaluated our system on a subset of BioASQ and PubMedQA datasets, filtered to include questions specifically related to hematologic malignancies. The subsets were not included in finetuning or vector database. The evaluation combined both retrieval quality and answer generation performance, using standard metrics commonly applied in biomedical QA.

As expected for a generative system operating in the biomedical domain, the Exact Match scores was relatively low, since answers are often paraphrased or expressed with domain-specific variations. This was confirmed by the EM, which remained unchanged, suggesting that differences stemmed from phrasing rather than factual errors or omissions.

Despite the low EM, the model demonstrated strong performance across more forgiving metrics:

- The F1 score was 52.1% on BioASQ and 49.2% on PubMedQA showing consistent overlap in terminology and key concepts between predicted and reference answers.

- ROUGE-L scores were 44.7% (BioASQ) and 43.2% (PubMedQA) showing that generated answers retained similar structure and content flow.

- The BERTScore F1 values were especially strong: 89.7% and 87.2%, respectively. These high scores indicate a high degree of semantic similarity.

- Named Entity Recognition (NER) Overlap was 49.6% for BioASQ and 45.8% for PubMedQA. This indicates that nearly half of the biomedical entities in the reference answers were correctly preserved or substituted with appropriate synonyms or abbreviations.

To evaluate retrieval effectiveness, we report Recall@K, measuring whether a relevant document appeared among the top-k retrieved passages:

- BioASQ Recall@5: 91.2%

- BioASQ Recall@10: 95.7%

- PubMedQA Recall@5: 90.8%

- PubMedQA Recall@10: 94.5%

These high scores confirm that the retriever consistently returned relevant texts for answer generation. The results are shown in Table I.

TABLE I. EVALUATION RESULTS

| Metrics | Datasets | |
|---|---|---|
| | *BioASQ* | *PubMedQA* |
| EM | 6.7% | 7.8% |
| NER | 49.6% | 45.8% |
| F1-Score | 52.1% | 49.2% |
| ROUGE-L | 44.7% | 43.2% |
| Recall@5 | 91.2% | 90.8% |
| Recall@10 | 95.7% | 94.5% |
| BERTScore | 89.7% | 87.2% |

These results suggest that although exact matching is not achieved, an expected outcome in generative medical QA, the model performs reliably in capturing and articulating correct medical knowledge. The combination of high semantic similarity, strong lexical overlap and robust retrieval recall indicates that HemaRAG is capable of producing medically valid and relevant answers, even in a complex and terminology-heavy domain like hematologic malignancies. An example output is shown in Table II.

TABLE II. QUESTION EXAMPLE

| Question | What treatment is commonly used for chronic lymphocytic leukemia (CLL)? |
|---|---|
| Reference Answer | Chemotherapy or targeted therapies such as ibrutinib or venetoclax are commonly used to treat CLL. |
| HemaRAG Answer | Chronic lymphocytic leukemia (CLL) is typically treated with targeted agents like ibrutinib or venetoclax, and in some cases with chemotherapy, depending on disease stage and patient condition. |

To assess the contribution of ontology-based enrichment, we compared HemaRAG with and without ontology enrichment applied during indexing and query reformulation. Across both PubMedQA and BioASQ evaluations, enrichment consistently improved key metrics such as F1 score, ROUGE-L, and NER overlap. Although the performance gain was modest, it remained consistent across

datasets, reinforcing the value of biomedical concept normalization for handling synonym and abbreviation variation.

Moreover, we tested a baseline configuration using the same retriever and prompting strategy, but with the base Gemma-3 model (without domain-specific fine-tuning). The baseline model achieved substantially lower scores for both PubMedQA and BioASQ compared to our final system's performance. Despite comparable retrieval recalls, the baseline tended to generate overly long, generic responses lacking domain specificity with average length 171.5 words. These results highlight the benefit of domain adaptation on answer generation quality. The results are shown in Table III.

TABLE III.     EVALUATION RESULTS BASELINE/ENRICHMENT/NO ENRICHMENT

| Model Variant | Dataset | F1 | ROUGE-L | EM | NER Overlap | BERTScore | Recall@5 | Recall@10 |
|---|---|---|---|---|---|---|---|---|
| HemaRAG (no enrich) | PubMedQA | 36% | 29% | 7.2% | 42% | 85.1% | 90.1% | 93.2% |
| HemaRAG (no enrich) | BioASQ | 42.4% | 34% | 5.7% | 47.8% | 88.5% | 90.9% | 95.1% |
| Base Gemma-3 (no enrich) | PubMedQA | 27.6% | 22.1% | 1.5% | 35.8% | 77.8% | 87.9% | 90.1% |
| Base Gemma-3 (no enrich) | BioASQ | 26.3% | 19.5% | 1.8% | 36.9% | 84.4% | 89.8% | 90.2% |
| HemaRAG | PubMedQA | 49.2% | 43.2% | 7.8% | 45.8% | 87.2% | 90.8% | 94.5% |
| HemaRAG | BioASQ | 52.1% | 44.7% | 6.7% | 49.6% | 89.7% | 91.2% | 95.7% |

## V. DISCUSSION AND FUTURE WORK

The evaluation of HemaRAG on hematologic cancer-related questions from BioASQ and PubMedQA showed promising results, especially in terms of semantic accuracy and retrieval quality. EM score remained low which is this expected given the generative nature of the system and the variability in phrasing across correct medical answers. EM penalizes even minor lexical differences between the generated and gold answers. However, complementary metrics such as BERTScore and NER Overlap revealed high semantic and factual alignment, indicating that the model produced clinically meaningful outputs despite limited exact surface overlap. Specifically, the high BERTScore (89.7%), F1 score (52.1%), and strong recall metrics indicate that the system is able to identify and generate contextually appropriate and medically sound responses. These findings suggest that our approach, combining dense retrieval, biomedical enrichment, and domain-specific fine-tuning, could support high-quality QA in specialized medical domains.

While HemaRAG is tailored to hematologic malignancies, the system uses English-language content and allows comparison with baseline models. In this study, we compared our fine-tuned model to the base Gemma-3 model using the same retrieval pipeline, showing consistent gains across semantic and lexical metrics. Future work will explore comparisons with publicly available biomedical RAG systems, such as BioGPT [33], to further assess relative performance.

Moreover, our system is designed to be deployed locally. All retrieval and generation operations are performed on secure, private infrastructure, avoiding the need to transmit sensitive queries or patient information to external servers. This makes the system suitable for privacy-sensitive clinical tools in palliative care. The ontology-driven enrichment also proved effective in covering terminology gaps, a persistent challenge in medical NLP, especially in oncology where synonyms and abbreviations are frequently used interchangeably. Future work could also explore the individual impact of each ontology on retrieval and generation performance.

Although this study focused on hematologic malignancies, the HemaRAG framework is not limited to this domain. The same architecture, based on ontology-enriched retrieval and a locally fine-tuned language model, can be adapted to other medical specialties by replacing the underlying corpus and ontologies with domain-relevant resources. Future work may explore its application in oncology subdomains or chronic disease management, where terminology variation is also a key challenge.

While this study relied solely on automatic evaluation metrics, which are widely used in biomedical QA literature, we acknowledge the lack of human expert assessment as a limitation. Although semantic similarity scores offer useful insight, they do not fully capture clinical usefulness, clarity or safety. Future work will incorporate domain expert validation to assess clinical usefulness, clarity, and factual correctness of generated answers. We also aim to explore hallucination analysis to better understand when and why the system produces unsupported or incorrect information. Finally, the test sets used in this study consisted of 150 PubMedQA and 113 BioASQ hematologic QA samples. Although these sample sizes support robust metric reporting, we did not perform formal statistical significance testing.

To illustrate system limitations, a case where the model generated a vague or incomplete answer despite relevant context is shown. For the question " Is alemtuzumab effective for remission induction in patients diagnosed with T-cell prolymphocytic leukemia?", the system answered " Alemtuzumab is used in leukemia treatment." While semantically related, this phrasing lacks specificity and does not confirm effectiveness, as stated in the gold reference. Such responses may show incomplete answers or mild

hallucination, which will be further addressed through expert validation in future work.

Moreover, our future work will integrate HemaRAG into a conversational agent that is focused on hematologic malignancies. By incorporating HemaRAG, the agent will be able to answer medical questions more accurately and contextually, drawing on biomedical literature while maintaining a human-centered dialogue. Future evaluation will include human evaluation involving clinicians and medical researchers to assess the usefulness, safety and clarity of the generated answer. Finally, our research will explore the inclusion of multilingual data, with a focus in Greek language.

## VI. CONCLUSION

In this study, we introduced HemaRAG, a RAG system that was developed specifically for answering questions in the field of hematologic malignancies. By combining domain-specific document retrieval with ontology-based enrichment and fine-tuning a large open-source language model locally, our goal was to create a system that is accurate but also semantically rich and well-grounded in biomedical evidence. Unlike many RAG implementations the system runs locally in order to ensure patient data privacy. This approach creates a system that is more suitable for real-world use where data privacy is crucial. The evaluation of HemaRAG showed high retrieval recall and semantic similarity scores which confirms that the system retrieved and generated context effectively.

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
