# OpenReview forum: "HemaRAG: A Retrieval-Augmented Generation System for Medical Question Answering in Hematologic Malignancies"
_IEEE.org/EMBS/BHI/2025/Conference — BHI 2025_

### Official Review · Reviewer_7Zez · 2025-06-29
**HemaRAG: A Retrieval-Augmented Generation System for Medical Question Answering in Hematologic Malignancies**

**Confidence:** 3
**Clarity Of Writing:** good
**Clinical Significance:** fair
**Methodological Novelty:** fair
**Overall Rating:** 4
**Final Rating:** 6

**Experiments And Results:**

fair

**Questions For The Authors:**

How many test questions were actually used in the evaluation, and are there confidence intervals or statistical significance tests for the reported metrics?
What is the baseline performance without ontology enrichment, and how much does each ontological resource (MeSH, NCIT, DOID, UMLS) contribute individually to system performance?
How does HemaRAG compare against other medical RAG systems or general-purpose LLMs on the same hematologic malignancy questions?
Are there examples of system failures or hallucinations, and what strategies are taken to address this?

**Strengths:**

The emphasis on local deployment and data privacy is highly relevant for clinical applications, distinguishing it from many RAG systems that rely on external APIs. The integration of multiple biomedical ontologies for synonym enrichment is a thoughtful approach to handling medical terminology variations. The two-stage fine-tuning strategy is methodologically sound. The evaluation uses appropriate metrics for biomedical QA. The system architecture is clearly described with good technical details about implementation choices like ChromaDB and quantization strategies.

**Summary Of The Paper:**

This paper presents HemaRAG specifically designed for answering medical questions about hematologic malignancies. The system combines a dense retriever enhanced with biomedical ontologies and a locally fine-tuned Gemma 3 27B model. The authors constructed a domain-specific corpus from PubMed abstracts and curated datasets, applying ontology-based enrichment to address terminology variations in medical literature. The system uses ChromaDB for local vector storage and retrieval to maintain data privacy. The entire system operates locally without external API calls, making it suitable for privacy-sensitive clinical environments.

**Weaknesses:**

The evaluation is limited to a small subset of questions without specification on the exact number or statistical significance testing, which is crucial with a small sample size. The paper lacks comparison with other medical RAG systems or domain-specific baselines, making it difficult to assess the relative performance. The extremely low exact match scores (6.7-7.8%) raise questions about practical utility, even though this is acknowledged as expected for generative systems. There's no clinical validation from medical experts to assess generated answers. The ontology enrichment process lacks systematic evaluation - it's unclear how much this component actually contributes to performance improvements. The fine-tuning details are insufficient, particularly regarding training data size, validation procedures, and hyperparameter selection. The paper doesn't address potential hallucination issues or provide error analysis of incorrect responses.

---

### Official Review · Reviewer_8BER · 2025-07-11
**A domain-specific, privacy-preserving RAG system for medical question answering in hematologic oncology.**

**Confidence:** 4
**Clarity Of Writing:** good
**Clinical Significance:** good
**Methodological Novelty:** good
**Overall Rating:** 7

**Experiments And Results:**

good

**Questions For The Authors:**

What types of errors or hallucinations were most common, and how might the system be improved to avoid them?

**Strengths:**

HemaRAG effectively addresses a real clinical need by focusing on medical question answering in the specialized domain of hematologic malignancies, an area underserved by current generalist models. Its fully local deployment ensures data privacy and makes it suitable for integration in clinical environments, particularly where patient confidentiality is critical. Moreover, the use of biomedical ontologies (MeSH, NCIT, UMLS, DOID) for enriching both the retrieval and query pipelines significantly improves semantic matching, helping the system handle synonym-rich and context-sensitive medical language with greater precision.

**Summary Of The Paper:**

HemaRAG is a thoughtfully designed RAG system that integrates ontology-enriched retrieval with a locally fine-tuned LLM to address medical QA in hematologic malignancies. It demonstrates strong technical performance, local deployability, and domain relevance, but would benefit from baseline comparisons and human expert evaluation to validate clinical utility.

**Weaknesses:**

The system lacks a direct comparison with strong general-purpose baselines such as GPT-4 or Med-PaLM, making it difficult to gauge the relative benefit of domain-specific fine-tuning. There is also no human evaluation or clinical expert feedback included, which limits our understanding of how safe, clear, or useful the generated answers are in real-world medical contexts.

---

### Official Review · Reviewer_LBFE · 2025-07-13
**Review for Submission Number: 307**

**Confidence:** 5
**Clarity Of Writing:** good
**Clinical Significance:** great
**Methodological Novelty:** fair
**Overall Rating:** 5
**Final Rating:** 6

**Experiments And Results:**

fair

**Questions For The Authors:**

1. Can the frmework presented in this work be generalized to other medical domains? Even some discussion around would be good to have to guide readers

2. It would also be helpful to add example breakdowns where the system fails or misleads.

3. Perhaps including ablation studies like RAG with vs without enrichment will help quantifying the added value of each pipeline component?

**Strengths:**

1. The use of MeSH, NCIT, DOID, UMLS for synonym normalization is a good approach to improve semantic retrieval and address common issues like: acronym ambiguity, term variation.

2. The paper mentions that all aspects of the pipeline is executed locally. So that is making the system well-suited for privacy-preserving clinical applications.

3. The paper reports multiple evaluation metrics (EM, F1, ROUGE-L, BERTScore, NER overlap, Recall@k) to provide a comprehensive assessment of both retrieval and generative components.

**Summary Of The Paper:**

This paper presents a RAG system designed for medical QA in hematologic malignancies. It includes a dense retriever (chromaBD) and a fine-tuned 27B Gemma LLM to answer medical queries. Evaluation is conducted on BioASQ and long-form PubMedQA benchmarks. Results: high semantic similarity (BERTScore ~87–89%), strong retrieval accuracy (Recall@10 ~95%), and meaningful lexical overlap (F1 ~50%), although Exact Match (EM) remains low due to natural language variability.

**Weaknesses:**

1. The paper cites previous studies suggesting that RAG helps reduce hallucinations. However, this work does not include a systematic analysis of hallucinations or any evaluation of confidence in the generated outputs to support that claim.

2. The paper refers to "ontology enrichment," but this term may be misleading, as it is unclear whether the proposed enrichment actually leads to improved RAG performance.

---

### Official Review · Reviewer_Dcox · 2025-07-18
**Ontology-Enriched Medical QA with Gaps in Evaluation**

**Confidence:** 3
**Clarity Of Writing:** good
**Clinical Significance:** good
**Methodological Novelty:** good
**Overall Rating:** 5
**Final Rating:** 6

**Experiments And Results:**

fair

**Questions For The Authors:**

1. Have you conducted any ablation studies to isolate the effect of MeSH/NCIT/DOID/UMLS enrichment on retrieval quality and answer generation? Without this, it’s unclear how critical that component is to your system.

2. Given the absence of human evaluation, how do you ensure that the generated answers are not only semantically aligned but also clinically accurate, safe, and useful in real-world scenarios?

**Strengths:**

The system is tailored for hematologic malignancies, combining specialized corpus curation with biomedical ontology enrichment. The design improves semantic retrieval in a terminology-heavy medical domain.

All components run locally with no external API dependencies, making it viable for clinical settings with strict privacy requirements.

The experiments show both semantic and lexical effectiveness on the tasks, supporting the customized designs.

**Summary Of The Paper:**

The paper proposed domain-specific Retrieval-Augmented Generation system designed for medical question answering in hematologic malignancies, using ontology-enriched dense retrieval with a fine-tuned local LLM. It achieved strong semantic similarity and high retrieval accuracy, while maintaining patient data privacy through fully local deployment. However, the system lacks human expert evaluation, generalizability beyond its narrow domain, and ablation studies to validate the impact of key components.

**Weaknesses:**

All evaluation relies on automatic metrics. There's no human validation, which weakens claims about real-world medical utility or safety.

EM scores remain poor (around 6.7–7.8%), which reflects model's difficulty in aligning generated outputs with gold-standard phrasing.

The proposed method lacks comparison with other RAG architectures, so the absolute gain from the proposed approach isn’t fully suppported.